# Genome-Wide Analysis of the Growth-Regulating Factor (GRF) Family in Aquatic Plants and Their Roles in the ABA-Induced Turion Formation of *Spirodela polyrhiza*

**DOI:** 10.3390/ijms231810485

**Published:** 2022-09-10

**Authors:** Gaojie Li, Yan Chen, Xuyao Zhao, Jingjing Yang, Xiaoyu Wang, Xiaozhe Li, Shiqi Hu, Hongwei Hou

**Affiliations:** 1The State Key Laboratory of Freshwater Ecology and Biotechnology, The Key Laboratory of Aquatic Biodiversity and Conservation of Chinese Academy of Sciences, Institute of Hydrobiology, Chinese Academy of Sciences, Wuhan 430072, China; 2University of Chinese Academy of Sciences, Beijing 100049, China; 3Zhejiang Marine Development Research Institute, Zhoushan 316021, China

**Keywords:** growth-regulating factors, aquatic plants, ABA, turion, *Spirodela polyrhiza*

## Abstract

Growth-regulating factors (GRFs) are plant-specific transcription factors that play essential roles in regulating plant growth and stress response. The *GRF* gene families have been described in several terrestrial plants, but a comprehensive analysis of these genes in diverse aquatic species has not been reported yet. In this study, we identified 130 *GRF* genes in 13 aquatic plants, including floating plants (*Azolla filiculoides*, *Wolffia australiana*, *Lemna minuta*, *Spirodela intermedia*, and *Spirodela polyrhiza*), floating-leaved plants (*Nymphaea colorata* and *Euryale ferox*), submersed plants (*Zostera marina*, *Ceratophyllum demersum*, *Aldrovanda vesiculosa*, and *Utricularia gibba*), an emergent plant (*Nelumbo nucifera*), and an amphibious plant (*Cladopus chinensis*). The gene structures, motifs, and cis-acting regulatory elements of these genes were analyzed. Phylogenetic analysis divided these *GRFs* into five clusters, and ABRE cis-elements were highly enriched in the promoter region of the *GRFs* in floating plants. We found that abscisic acid (ABA) is efficient at inducing the turion of *Spirodela polyrhiza* (giant duckweed), accompanied by the fluctuated expression of *SpGRF* genes in their fronds. Our results provide information about the *GRF* gene family in aquatic species and lay the foundation for future studies on the functions of these genes.

## 1. Introduction

Growth-regulating factors (GRFs) are plant-specific transcription factors, which regulate cell proliferation and expansion and play important roles in plant development and environmental response [1,2]. The GRF family proteins contain the following two conserved regions: the QLQ and the WRC domains [3]. The QLQ domain functions as a protein–protein interaction domain, which interacts with the conserved structure of other proteins to perform transcriptional activation [4,5]. The WRC domain contains a nuclear localization signal (NLS) motif and regulates the expression of downstream genes by binding to their cis-acting regions [6]. The *OsGRF1* was the first identified *GRF* from *Oryza sativa*, which participates in the gibberellin (GA)-induced stem elongation [7]. In *A. thaliana*, GRFs can form complexes with GRF-interacting factors (GIFs) and control leaf size and longevity during the progression of leaf development [8]. In addition, GRFs also regulate the processes of floral organogenesis [9], root development [10], and grain size [11]. The microRNA miR396 directly inhibits the expression of *GRFs* through post-transcriptional regulation and participates in the regulation of leaf expansion and other processes [2,12]. Subsequently, genome-wide identifications were performed on diverse plant species, such as *Arabidopsis thaliana* [13], *Nicotiana tabacum* [14], *Triticum aestivum* [15], *Glycine max* [16], and *Actinidia chinensis* [17]. *GRFs* are regulated by many phytohormones (auxin, cytokinin (CK), abscisic acid (ABA), GA, etc.) and function in a manner that is responsive to hormone signal transduction [14]. Recent research also showed that *GRFs* are involved in the plant’s adaptation to many abiotic stresses, such as shade [16], salinity [18], cold, and drought [19].

Aquatic plants are one of the important components of aquatic ecosystems, as well as important producers of oxygen and energy in the water, and some of them are hyperaccumulators for the bioaccumulation of heavy metals in aquatic conditions [20,21,22]. Although long-neglected, these plants have valuable scientific research and applied uses, and their unique environmental adaptability provides essential insights for studies on plant adaption [23]. Recently, a study has shown that *GRFs* are candidate differentially expressed genes (DEGs) involved in the phenotypic plasticity of the aquatic plant *Potamogeton octandru* [24], indicating the important roles of *GRFs* in aquatic plants. However, though the characteristics of *GRFs* have been widely described in terrestrial species, the genome-wide identification and analysis of the GRF families in aquatic plants are still lacking. Due to the important functions of GRFs in plant adaptation, it is urgent to identify GRFs in aquatic species, which may contribute to a better understanding of the molecular mechanisms of adaptation to the submerged environment in aquatic plants.

Despite the general trend to live on dry land, several ancestors of aquatic plants have ventured back into the freshwater regimes or even marine habitats and developed into modern aquatic plants [25]. It is known that ABA signal pathways and transduction are conserved throughout the evolution of basal plants to angiosperms and function as important regulators for plant growth and stress response in many terrestrial species [26]. Furthermore, ABA was recently verified to be involved in the phenotypic plasticity of some aquatic plants, regulating their leaf shapes and stomata development and participating in their response to environmental changes [27,28,29]. Therefore, ABA seems to be an important regulator of the development and responses of aquatic plants to the surrounding environments. However, it is still unknown whether it plays an essential role in diverse aquatic species.

In the present study, a genome-wide identification and comprehensive analysis of the GRF family members in 13 aquatic plants was conducted. Their sequence characteristics, phylogenetic relationships, gene structures, conserved motif compositions, and cis-acting regulatory elements were then characterized. The chromosomal localization, gene duplication, and synteny of the *GRF* genes in *Spirodela polyrhiza* (giant duckweed) were further analyzed as a representative. Based on our results, we found that the ABRE cis-elements were highly enriched in the promoter region of the *GRFs* in floating plants. ABA is thought to be the trigger responsible for inducing a response to environmental stresses, and it is efficient to induce turion in duckweeds [30]. Therefore, ABA may regulate genes involved in this process, probably *GRFs*. We then analyzed the expression pattern of *SpGRFs* during the ABA-induced turion formation of *S. polyrhiza*. This study provides an essential foundation for the studies of the GRF family members in a wide range of aquatic plants, especially in the turion formation of duckweeds.

## 2. Results

### 2.1. Identification and Phylogenetic Analysis of GRF Genes in 13 Aquatic Plants

Based on the BLASTp of 9 GRFs from *A. thaliana* and the Hidden Markov Model (HMM) of the WRC (PF08879) and QLQ (PF08880) domains, a total of 130 candidate GRFs were obtained from 13 aquatic plants, including 5 floating plants (*Azolla filiculoides*, *Wolffia australiana*, *Lemna minuta*, *Spirodela intermedia*, and *Spirodela polyrhiza*), 2 floating-leaved plants (*Nymphaea colorata* and *Euryale ferox*), 1 emergent plant (*Nelumbo nucifera*), 4 submersed plants (*Zostera marina*, *Ceratophyllum demersum*, *Aldrovanda vesiculosa*, and *Utricularia gibba*), and 1 amphibious plant (*Cladopus chinensis*). We found that the submerged plant *A. vesiculosa* contains a large number of GRFs, while the fern *A. filiculoides* and the duckweed *L. minuta* have only three GRFs (Figure 1). Subsequently, the characteristics of these GRFs, including protein length, isoelectric point (pI), molecular weight (MW), NLS type, and putative subcellular localization, were analyzed (Appendix A). As a result, the predicted protein lengths of these GRFs varied from 139 to 818 aa. We also found that the pI value ranges were 5.32–10.88 kDa, and the corresponding MWs of these GRFs were 15.85–87.31 kDa. The results of the NLS analysis showed that most of the NLSs of these GRFs are predicted to be bipartite NLSs, and some of them are predicted to be both bipartite and monopartite NLSs (AvGRF14, LmGRF3, SpGRF5, UgGRF2, WaGRF1, ZmGRF3, ZmGRF4, and ZmGRF10), and only the NLS of UgGRF1 is predicted to be a monopartite NLS (Appendix A). Furthermore, a subcellular location analysis showed that most of the predicted GRF proteins from these aquatic plants were putatively located in the nucleus, and a few proteins were putatively located in the cell membrane, cell wall, and chloroplast (Appendix A).

The sequence-based phylogenetic analysis among these aquatic plants and *A. thaliana* showed that these proteins could be clarified into five distinct groups (A–E, Figure 2). The largest cluster was group A, with 59 members from all 14 species (42.44%). Group B had 41 members from all 14 species (29.50%); group C had 20 members from 9 species (14.39%); group D had 13 members from 7 species (9.35%); the smallest cluster was group E, with 6 members from *A. thaliana*, *A. vesiculosa*, *C. chinensis*, and *U. gibba* (4.32%) (Figure 2).

### 2.2. Gene Structure and Conserved Motifs of the GRF Genes

The GSDS online server was used to perform the gene structures of all 139 *GRFs*. We found that most genes in the same group of the phylogenetic tree had similar gene structures (Figure 3). Among those, *EfGRF9* and *CdGRF7* had the largest intron length (approximately 12 kb), with seven or five exons, respectively. We also analyzed the conservative structure of these *GRFs*. Five representative motifs were analyzed through the MEME program, and their positions were illustrated on each gene (Figure 4). We found that all GRFs contain motifs 1 and 2. Group A contains motifs 1, 2, 3, and 4 in most members; group B contains all five motifs in most members; group C contains motifs 1, 2, 3, and 5 in most members; group D contains motifs 1, 2, and 4 in most members; group E contains motifs 1, 2, and 5 in most members. The characteristics of these motifs are listed in Appendix A.

### 2.3. Cis-Regulatory Element Analysis of the GRF Genes

The 1.5 kb upstream regions of the GRF genes in *A. thaliana* and 13 aquatic plants were extracted from genome databases to analyze their cis-acting elements (Figure 4). Several types of cis-acting elements were identified through the online software PlantCARE, and we identified elements related to plant development (GCN4_motif, CAT-box, circadian, and O_2_-site) and phytohormone responses, such as the MeJA-responsive (CGTCA-motif), the abscisic acid-responsive (ABRE), the salicylic acid responsiveness (TCA-element), the auxin-responsive (TGA-element), the gibberellin-responsive (GARE-motif), and the ethylene-responsive (ERE). We also identified elements related to abiotic stress responses (anaerobic induction (ARE), low-temperature responsive (LTR), MYB binding site involved in drought inducibility (MBS), defense, and stress responsiveness (TC-rich repeats)). Among those, we found that the ABRE cis-elements were highly enriched in the promoter region of the *GRFs* in the floating plants (Figure 5).

### 2.4. Chromosomal Localization, Gene Duplication, and Synteny Analysis of the GRFs in S. polyrhiza

Subsequently, we performed a synteny analysis on *S. polyrhiza*, which is a representative of the floating plants. Six *GRFs* were identified in *S. polyrhiza*, and their chromosome distributions were then analyzed. SpGRF1 (Spo001498) was located on chromosome number 1; SpGRF2 (Spo004459) and SpGRF3 (Spo005290) were located on chromosome number 3; SpGRF4 (Spo012722) was located on chromosome number 11; SpGRF5 (Spo013751) was located on chromosome number 14; SpGRF6 (Spo015563) was located on chromosome number 18 (Figure 6A). In addition, we found that no tandem repeat or segmental duplication events of *SpGRFs* occurred in *S. polyrhiza*, while *SpGRF1* and *SpGRF3* had collinearity (Figure 6B). The evolutionary relationships of the *GRFs* between *S. polyrhiza* and four representative species, including *N. nucifera* (Nelumbonaceae), *N. colorata* (Nymphaeaceae), *C. chinensis* (Podostemaceae), and *A. thaliana* (Brassicaceae), were further analyzed. Five syntenic *GRF* gene pairs were identified between *S. polyrhiza* and *N. nucifera*; four syntenic *GRF* gene pairs were identified between *S. polyrhiza* and *N. colorata*; two syntenic *GRF* gene pairs were identified between *S. polyrhiza* and *C. chinensis*; three syntenic *GRF* gene pairs were identified between *S. polyrhiza* and *A. thaliana* (Figure 7).

### 2.5. Expression Analysis of the SpGRF Genes in the Process of ABA-Induced Turion Formation

The turions of *S. polyrhiza* can be distinguished from normal fronds by their smaller size and thicker cell walls, which are used as a strategy to avoid environmental stresses [31]. To analyze the function of the *SpGRFs* in the turion formation of *S. polyrhiza*, we first performed a qRT-PCR to detect their relative expressions in the frond and turion of *S. polyrhiza*. We found that the *SpGRF3* was significantly upregulated in the turion (Figure 8). It was reported that ABA is efficient at inducing turions of *S. polyrhiza* [30]. Furthermore, we detected the expression patterns of all six GRFs during the ABA-induced turion formation and found that the expressions of most *SpGRFs* (except for *SpGRF4*) were fluctuant. The expression trends of most *SpGRFs* decreased under 3 days of treatment, then gradually increased, and finally decreased after 14 days of treatment in this process (Figure 9).

## 3. Discussion

GRFs are plant-specific transcription factors that regulate plant morphogenesis and stress resistance [32,33]. In the current study, a total of 130 *GRF* genes were identified in 13 aquatic plants, including floating plants, floating-leaved plants, emergent plants, submersed plants, and amphibious plants. We found that the numbers of GRFs in these aquatic plants seem to have gradually increased during species differentiation, as the copy numbers of the GRFs in eudicots are higher than in most ferns and monocots (Figure 1), which is similar to the discovery in land plants [18]. Previous studies have predicted [17,34] or verified [35] the subcellular localization of some GRFs, and many GRFs are located in the nucleus. However, it is still unknown which type of NLS is present in these GRFs. Here, we found that though most of the GRFs in these aquatic plants have bipartite NLSs, they can have either monopartite or bipartite NLSs and are not specific to only one type. These different NLSs may have diverse biological functions or unique evolutionary significance. Furthermore, we analyzed motifs in these *GRFs* and found that genes containing different motifs can be clarified into distinct groups. Motif 1 was associated with the WRC domain, and motif 2 was associated with the QLQ domain. Phylogenetic tree analysis revealed that these *GRF* members of the 13 aquatic species belong to five big groups, and genes in each group have similar gene structures.

Unlike most animals that can escape from stressful conditions, plants are sessile organisms that must undergo many environmental stresses directly at the place where they germinated. Therefore, plants have evolved many strategies for their adaptation, and cis-acting elements in the gene promoters were verified to be quite efficient to regulate their response to environmental changes [36]. The relative homogeneity of the water environment, the cloning propagation of aquatic plants, the diverse and efficient transmission modes, and the strong phenotypic plasticity are important factors for the wide distribution of aquatic plants [37]. However, no comprehensive analysis of the cis-acting elements has been performed on many aquatic species. Therefore, we analyzed 1.5 kb upstream regions of the *GRF* genes in 13 aquatic plants and identified elements related to plant development (GCN4_motif, CAT-box, circadian, and O2-site) and phytohormone response (MeJA/ABA/salicylic acid/auxin/GA/ethylene-responsive), and identified elements related to abiotic stress responses, such as ARE, LTR, MBS, and TC-rich repeats. Among those, we found that the ABRE cis-elements were highly enriched in the promoter region of the GRFs in floating plants (Figure 4), suggesting the possible function of the ABA in the regulation of *GRFs* in floating plants.

It was reported that *GRFs* play an important role in the response to environmental stresses [38]. In *A. thaliana*, *GRF5* regulates genes related to cold responses [33], and *GRF7* is responsible for salinity and drought responses [39]. In *Medicago truncatula*, *MtGRF5* contains more ABRE elements than other genes and responds effectively to osmotic stress [34]. Subsequently, we detected gene duplication and divergence in *S. polyrhiza*, a representative of floating plants. We found no tandem repeat or segmental duplication events among the six *SpGRFs*, while *SpGRF1* and *SpGRF3* had collinearity. Collinearity analyses between species showed that five syntenic *GRF* gene pairs were identified between *S. polyrhiza* and *N. nucifera*; four syntenic *GRF* gene pairs were identified between *S. polyrhiza* and *N. colorata*; two syntenic *GRF* gene pairs were identified between *S. polyrhiza* and *C. chinensis*; three syntenic *GRF* gene pairs were identified between *S. polyrhiza* and *A. thaliana*. We found that *SpGRF6* was collinear with other *GRF* genes in these four plants, indicating that *SpGRF6* and its orthologs were highly conserved among these species.

In many terrestrial plants, ABA is produced under osmotic stress and functions as a key regulator in the abiotic stress responses and tolerance of diverse plants [40]. Recently, the function of ABA was verified to be not only conserved in all terrestrial macrophytes but also conserved in many aquatic plants from ferns to angiosperms [25,41]. Previously, studies have shown that ABA was efficient at inducing turion formation in *S. polyrhiza*; turions are special organs that decrease in size and increase thickness for resistance to abiotic stresses [42,43], suggesting the important role of ABA in the adaptation of *S. polyrhiza*. In *A. thaliana*, the leaves grow through cell proliferation and cell expansion. *GRF5* is a positive regulator of leaf development in *A. thaliana*, whose overexpression results in the formation of larger leaves [44]. In contrast, *GRF9* is a negative regulator of leaf growth since the overexpression of *GRF9* decreases organ size and the *grf9* mutant produces bigger rosette leaves in *A. thaliana* [45]. Here, we found that *SpGRF3* was significantly upregulated in the turion when compared to the frond, while the expression in the frond decreased during the ABA-induced turion formation. These results suggest that *SpGRF3* may be a negative regulator of leaf size in *S. polyrhiza*, which needs to be further investigated in the future.

## 4. Materials and Methods

### 4.1. Identification of the GRF Genes

The genome databases and protein databases of *Wolffia australiana*, *Lemna minuta*, *Spirodela intermedia*, *Spirodela polyrhiza*, *Nymphaea colorata*, *Euryale ferox*, *Nelumbo nucifera*, *Zostera marina*, *Ceratophyllum demersum*, *Aldrovanda vesiculosa*, *Utricularia gibba*, and *Cladopus chinensis* were downloaded from the NCBI database (https://www.ncbi.nlm.nih.gov/, accessed on 28 February 2021), the Phytozome database (https://phytozome.jgi.doe.gov/pz/, accessed on 28 February 2021), the CoGe database (https://genomevolution.org/coge/, accessed on 28 February 2021), and the Ensembl plant database (https://plants.ensembl.org/index.html, accessed on 28 February 2021). The genome database of *Azolla filiculoides* was downloaded from the Fernbase (https://www.fernbase.org/, accessed on 12 February 2021). The Hidden Markov Model (HMM) profiles of the WRC (PF08879) and the QLQ (PF08880) were downloaded from the Pfam database (http://pfam.xfam.org/, accessed on 1 March 2021), and we also used HMMER (http://hmmer.org/, accessed on 31 August 2022) and InterPro (https://www.ebi.ac.uk/interpro/, accessed on 31 August 2022) for the identification. All of the GRF sequences of *A. thaliana* were downloaded from the NCBI website and used as query sequences to search for GRFs in the other 13 aquatic species by BLASTp with an E-value cutoff set as 1 × 10^−5^. The putative GRF genes were further confirmed by the SMART database (http: //smart.embl-heidelberg.de/, accessed on 23 April 2021) and the NCBI Conserved Domain database (http://www.ncbi.nlm.nih.gov/Structure/cdd/wrpsb.cgi, accessed on 23 April 2021).

### 4.2. Physiochemical Properties and Subcellular Localization

The protein length, isoelectric point (pI), and molecular weight (MW) of the GRF proteins were calculated using the online ExPASy-ProtParam (http://web.expasy.org/protparam/, accessed on 28 February 2021). The NLSs of all GRFs were predicted by the NLS Mapper (https://nls-mapper.iab.keio.ac.jp/cgi-bin/NLS_Mapper_form.cgi/, accessed on 31 August 2022) [46] with a cut-off score set as 4.0 and entire regions searched for bipartite NLSs with a long linker. The putative subcellular localization of all GRF proteins was predicted by the Cello v2.5 software (http://cello.life.nctu.edu.tw/, accessed on 28 February 2021).

### 4.3. Phylogenetic Classification, Gene Structures, and Motif Analysis

A neighbor-joining phylogenetic tree of all identified GRFs was constructed in MEGA 7.0 (http://www.megasoftware.net/, accessed on 28 April 2021) with 1000 bootstrap replicates on the JTT model, and the tree was displayed using the online software iTOl (https://itol.embl.de/, accessed on 28 April 2021). The Gene Structure Display Server 2.0 online program (GSDS 2.0, http://gsds.gao--lab.org/, accessed on 28 April 2021) was used to analyze the exon–intron structure information of all *GRF* genes. The MEME online program was used to identify the conserved motifs in the GRF proteins (https://meme-suite.org/meme/tools/meme, accessed on 28 April 2021).

### 4.4. Cis-Acting Elements, Chromosomal Localization, Gene Duplication, and Synteny Analysis

The 1.5 kb promoter sequence of all *GRF* genes was extracted from the genome databases using the TBtools software (V1.9832.0.0) [47]. The cis-acting elements of the promoter sequence were analyzed with PLANTCARE (http://bioinformatics.psb.ugent.be/webtools/plantcare/html/, accessed on 28 April 2021) and visualized with TBtools. The chromosomal distribution of the *SpGRF* genes was determined from the annotation file (GFF3) and visualized with TBtools. The synteny and gene duplication of the *SpGRF* genes were analyzed with MCScanX [48] and TBtools. The syntenic relationships among *S. polyrhiza* and other representative species were also identified with MCScanX and visualized with TBtools.

### 4.5. Expression Analysis of the SpGRF Genes by qRT-PCR (Quantitative Real-Time PCR)

Ten fronds of *S. polyrhiza* 7498 were aseptically transplanted into a half-strength Schenk and Hildebrandt basal salt mixture (Sigma, S6765) with a 1% sucrose liquid medium at pH 5.8. The cultures were kept in a growth chamber maintained at 60 μmol m^−2^ s^−1^ and 25 °C through a 16-h light and 8-h dark photoperiod. A total of 1 μM of ABA (Sigma) was added to each group, as previously reported [30]. One gram of fresh fronds was taken from a time course of 0 (no ABA), 3, 7, 10, and 14 days of the ABA treatment and frozen in liquid nitrogen. For each time point, we used three biological replicates. The high-quality total RNA was then extracted using a commercial RNA extraction kit (CWBIO), and cDNA was synthesized from 1 μg of the total RNA using a RT reagent Kit (Takara). The TB Green^®^ Premix Ex Taq^™^ Kit (TaKaRa, Dalian, China) and the Bio-Rad CFX96 touch real-time PCR system (Bio-Rad, Hercules, CA, USA) were used to perform a qRT-PCR. The qRT-PCR program was run under the following conditions: 10 min at 95 °C, 40 cycles of 95 °C for 15 s, 57 °C for 30 s, and 72 °C for 30 s. The Actin gene was used as an internal control [49]. The 2^−ΔΔCt^ method was employed to calculate the relative expression of the *SpGRF* genes. The data were presented as means ± standard deviations. A one-way analysis of variance (ANOVA) and the Duncan test were applied to determine the significant differences at the *p* < 0.05 level. All primers used for the qRT-PCR are listed in Appendix A.

## 5. Conclusions

This study analyzed the *GRF* genes in 13 aquatic plants through a genome-wide identification. A total of 130 *GRFs* were identified, and these genes show diverse amino acid lengths, motif compositions, and gene structures. In addition, the phylogenetic and collinearity analyses on the GRFs in the different species revealed their evolutionary patterns. The ABRE cis-elements were highly enriched in the promoter region of the *GRFs* in floating plants. Furthermore, the expression levels of *SpGRF1*, *SpGRF3*, and *SpGRF5* are significantly different in the frond and turion of *S. polyrhiza*. It was found that most genes (except for *SpGRF4*) were wavily changed and finally decreased in fronds during the ABA-induced turion formation; they may play an important role in the response to abiotic stresses. This study revealed the characteristics of the *GRF* family genes in several aquatic plants, which provide insights into understanding the evolutionary relationship and gene functions of aquatic species in diverse statuses.

## Figures and Tables

**Figure 1 ijms-23-10485-f001:**
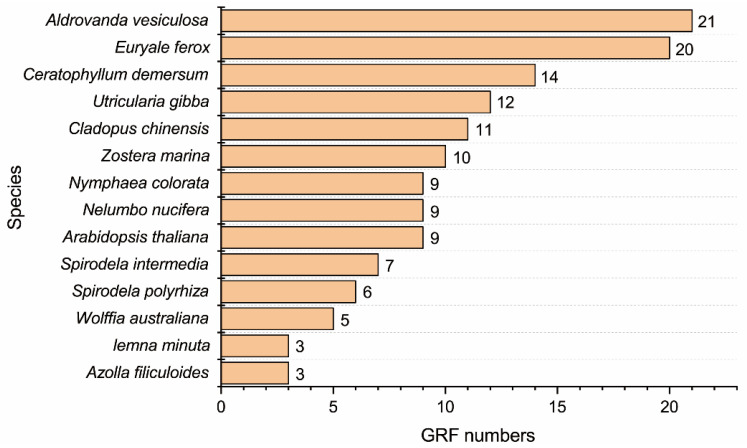
Calculations of the GRFs in *A. thaliana* and 13 aquatic plants.

**Figure 2 ijms-23-10485-f002:**
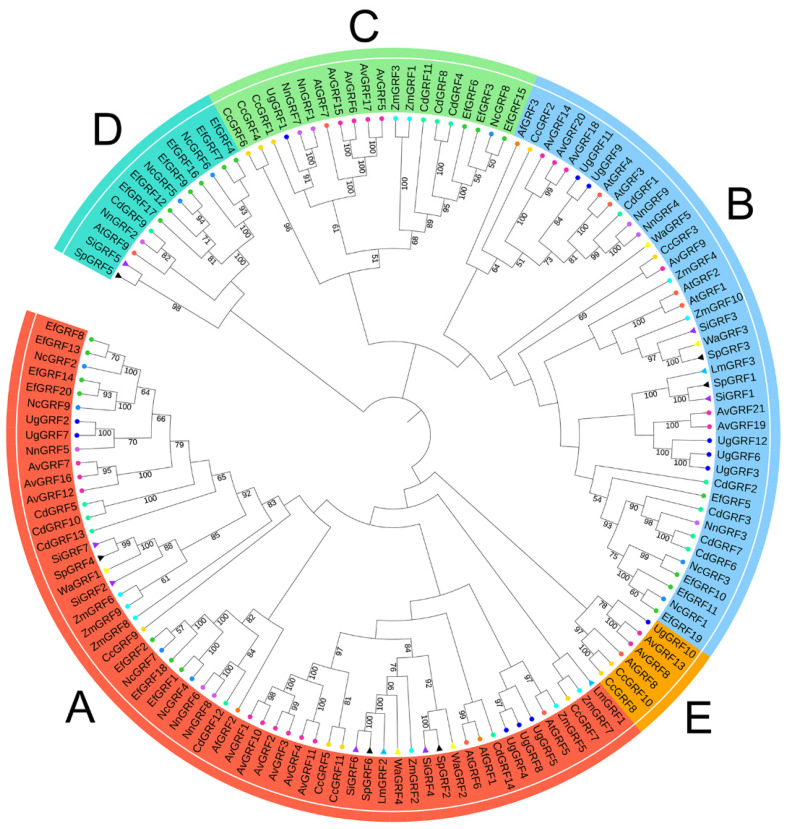
Phylogenetic analysis of the GRFs in *A. thaliana* and 13 aquatic plants. The neighbor-joining (NJ) phylogenetic tree was generated by MEGA 7.0 with 1000 bootstrap replications on the JTT model. The 139 GRFs were divided into five groups (**A**–**E** named by numbers of genes and indicated by five different colors), and the GRFs from different species were labeled with different colored symbols. Tomato circles indicate genes from *A. thaliana*; Dark orange circles indicate genes from *A. filiculoides*; Maroon circles indicate genes from *A. vesiculosa*; Gold circles indicate genes from *C. chinensis*; Yellow circles indicate genes from *C. demersum*; Dark gold circles indicate genes from *E. ferox*; Blue circles indicate genes from *U. gibba*; Dodger blue circles indicate genes from *N. colorata*; Medium orchid circles indicate genes from *N. nucifera*; Cyan circles indicate genes from *Z. marina*; Deep sky blue triangles indicate genes from *L. minuta*; Purple triangles indicate genes from *S. intermedia*; Black triangles indicate genes from *S. polyrhiza*; Yellow triangles indicate genes from *W. australiana*.

**Figure 3 ijms-23-10485-f003:**
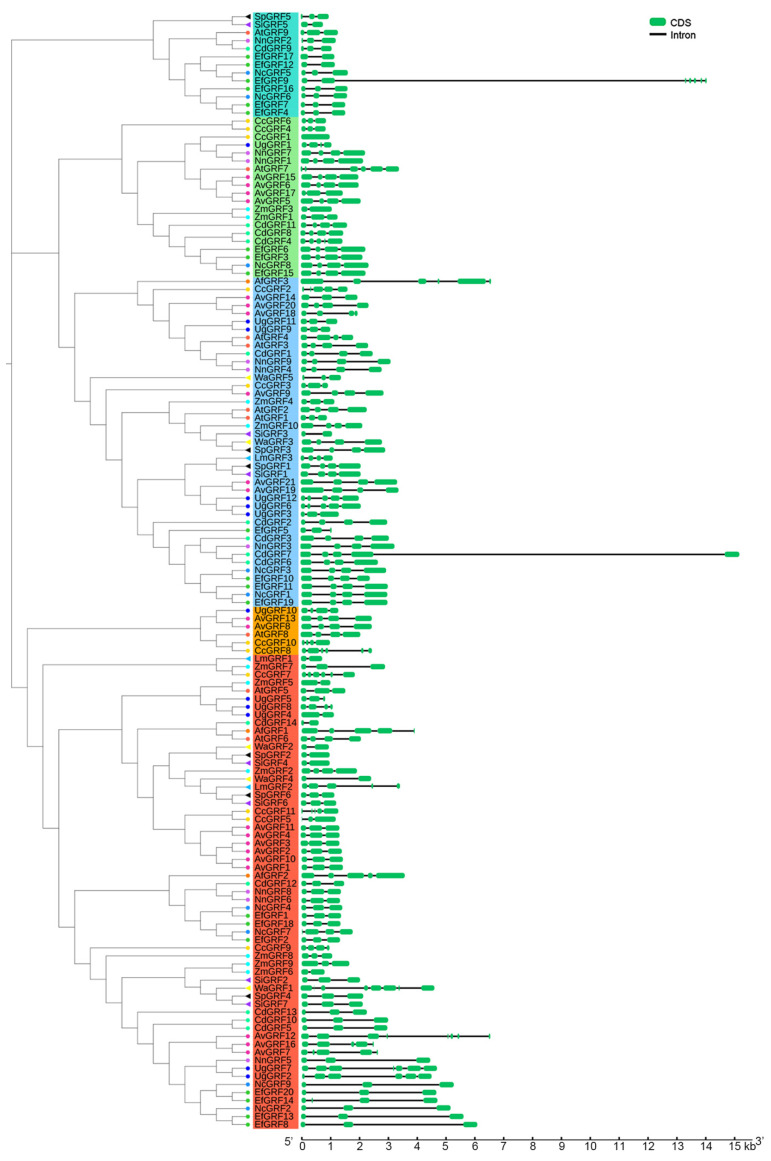
Gene structures of the *GRFs*. The GRFs from different species were labeled with different colored symbols. Tomato circles indicate genes from *A. thaliana*; Dark orange circles indicate genes from *A. filiculoides*; Maroon circles indicate genes from *A. vesiculosa*; Gold circles indicate genes from *C. chinensis*; Yellow circles indicate genes from *C. demersum*; Dark gold circles indicate genes from *E. ferox*; Blue circles indicate genes from *U. gibba*; Dodger blue circles indicate genes from *N. colorata*; Medium orchid circles indicate genes from *N. nucifera*; Cyan circles indicate genes from *Z. marina*; Deep sky blue triangles indicate genes from *L. minuta*; Purple triangles indicate genes from *S. intermedia*; Black triangles indicate genes from *S. polyrhiza*; Yellow triangles indicate genes from *W. australiana*.

**Figure 4 ijms-23-10485-f004:**
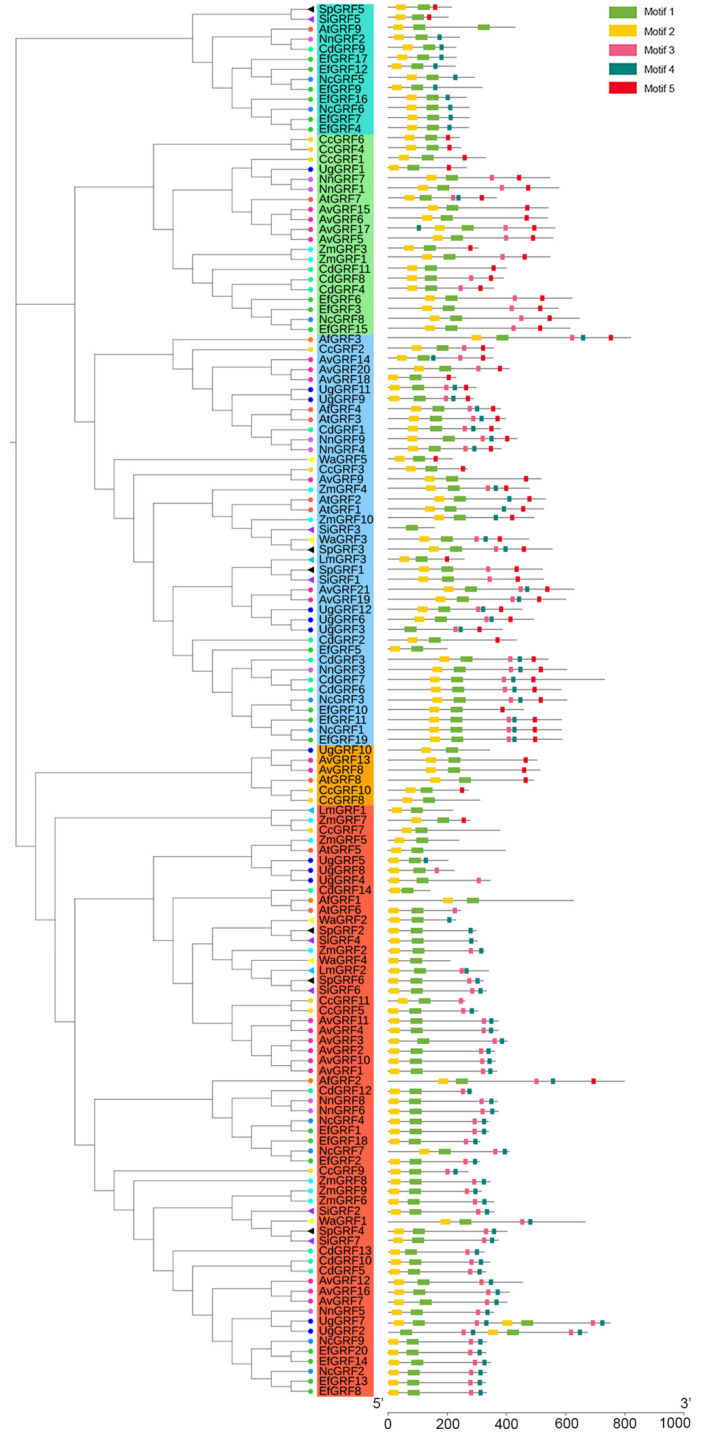
Conserved motifs of the *GRFs*. The GRFs from different species were labeled with different colored symbols. Tomato circles indicate genes from *A. thaliana*; Dark orange circles indicate genes from *A. filiculoides*; Maroon circles indicate genes from *A. vesiculosa*; Gold circles indicate genes from *C. chinensis*; Yellow circles indicate genes from *C. demersum*; Dark gold circles indicate genes from *E. ferox*; Blue circles indicate genes from *U. gibba*; Dodger blue circles indicate genes from *N. colorata*; Medium orchid circles indicate genes from *N. nucifera*; Cyan circles indicate genes from *Z. marina*; Deep sky blue triangles indicate genes from *L. minuta*; Purple triangles indicate genes from *S. intermedia*; Black triangles indicate genes from *S. polyrhiza*; Yellow triangles indicate genes from *W. australiana*.

**Figure 5 ijms-23-10485-f005:**
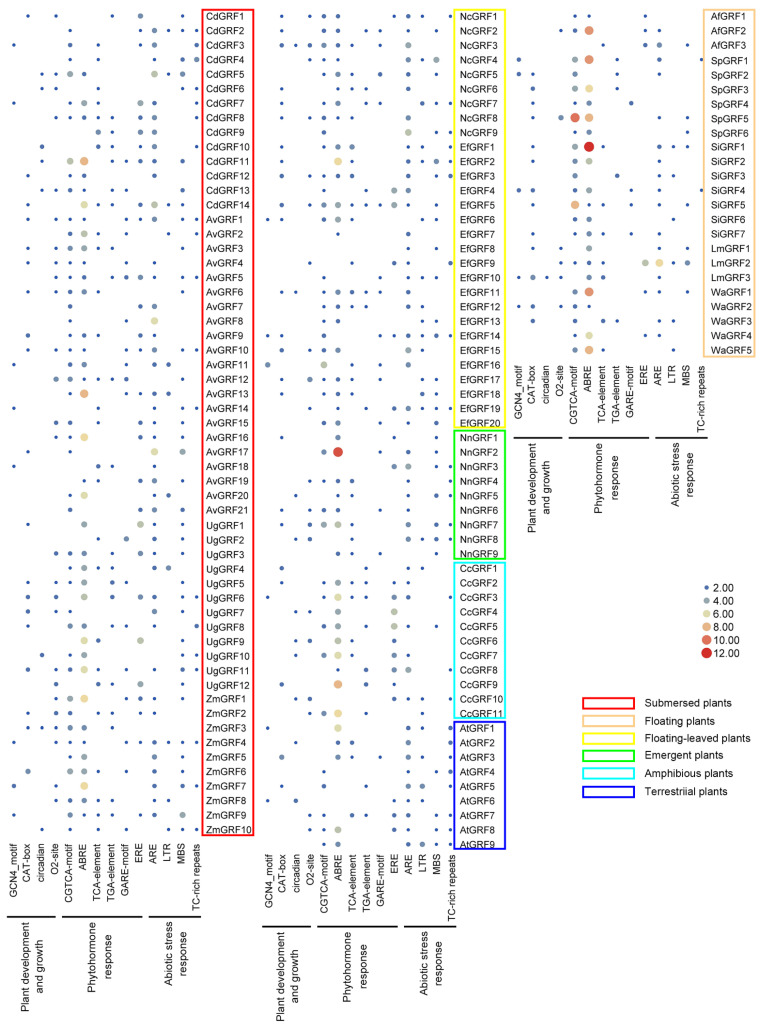
Frequency of the main cis-elements in the 1.5 kb promoter of the 139 *GRFs*.

**Figure 6 ijms-23-10485-f006:**
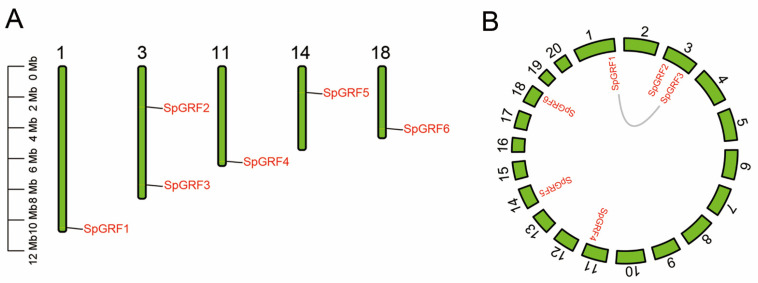
Chromosomal localization, gene collinearity, and synteny analysis of the *SpGRFs*. (**A**) Distribution of the *SpGRF* genes on the chromosomes of *S. polyrhiza*. The scale bar indicates chromosome length (Mb). (**B**) Gene collinearity of the *SpGRF* genes on the chromosomes of *S. polyrhiza*.

**Figure 7 ijms-23-10485-f007:**
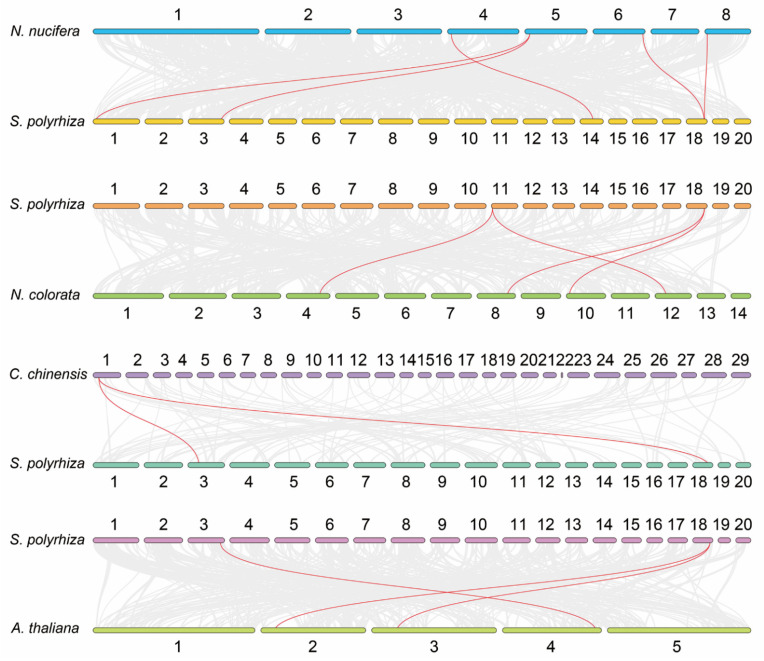
Synteny relationship of the *GRF* gene pairs between *S. polyrhiza* and four other plants. The red colour represents syntenic *GRFs*, and the gray lines show the collinear blocks of the plant genome. The chromosome number was labeled near each chromosome.

**Figure 8 ijms-23-10485-f008:**
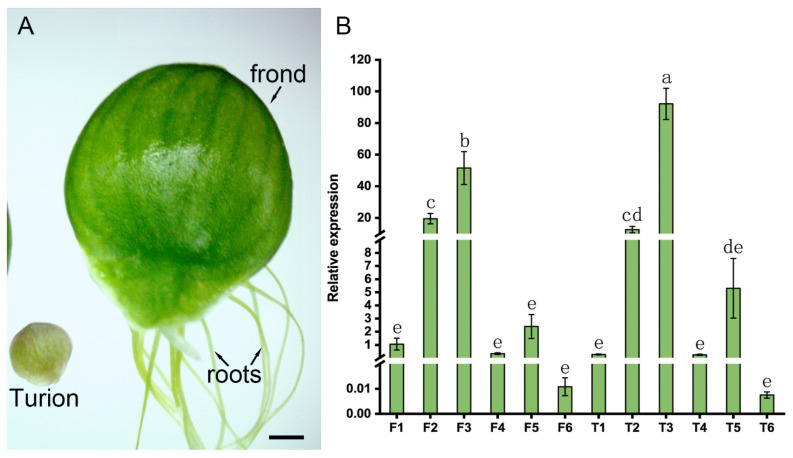
Phenotypes and expression of the *SpGRFs* in the frond and turion of *S. polyrhiza*. (**A**) Phenotypes of turion (left) and whole plant (right) of *S. polyrhiza*. Bar = 1 mm. Arrows indicate frond or roots of *S. polyrhiza*, separately. (**B**) Expression of the *SpGRFs* in the frond and turion. F1: *SpGRF1* in the frond; F2: *SpGRF2* in the frond; F3: *SpGRF3* in the frond; F4: *SpGRF4* in the frond; F5: *SpGRF5* in the frond; F6: *SpGRF6* in the frond; T1: *SpGRF1* in the turion; T2: *SpGRF2* in the turion; T3: *SpGRF3* in the turion; T4: *SpGRF4* in the turion; T5: *SpGRF5* in the turion; T6: *SpGRF6* in the turion. The different letters indicate statistically significant differences relative to F1, as determined by the Duncan test (*p* < 0.05).

**Figure 9 ijms-23-10485-f009:**
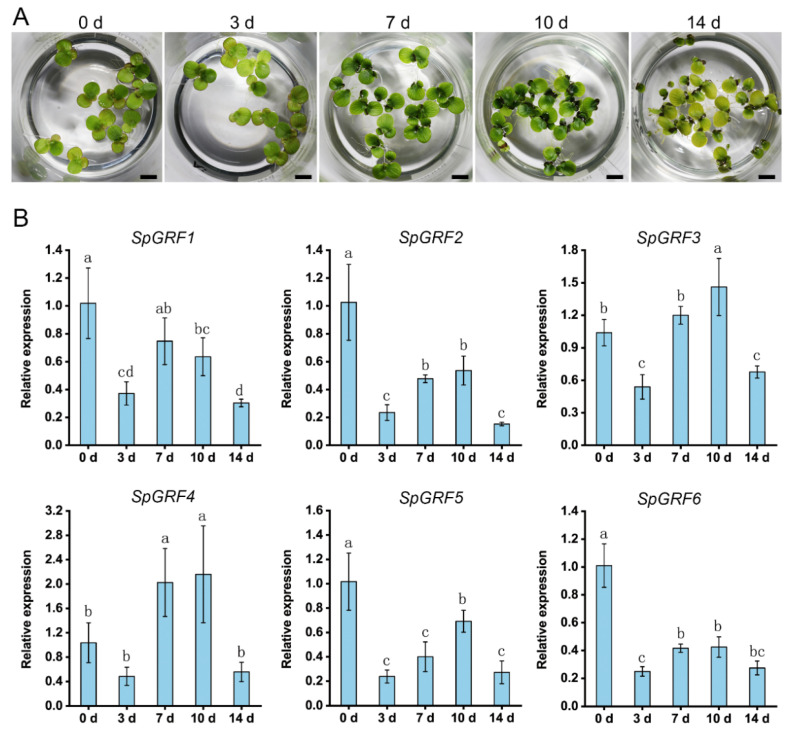
Phenotypes and expression of the *SpGRFs* during the ABA-induced turion formation of *S. polyrhiza*. (**A**) Phenotypes of *S. polyrhiza* during the ABA-induced turion formation. Bars = 1 cm. (**B**) Expression of the *SpGRFs* in the fronds after the treatment. The different letters indicate statistically significant differences relative to 0 days, as determined by the Duncan test (*p* < 0.05).

## Data Availability

The datasets used and/or analyzed during the current study are available from the corresponding author on reasonable request.

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
