# Peer review of "Genome-Wide Analysis of the Growth-Regulating Factor (GRF) Family in Aquatic Plants and Their Roles in the ABA-Induced Turion Formation of Spirodela polyrhiza"

_ijms, 2022, doi:10.3390/ijms231810485_

Round 1

Reviewer 1 Report

The premise and the aim of this paper to characterize the GRF genes of aquatic plants is something that I can say quite new for gene characterization study in general. However, some dictions added to this manuscript are quite confusing. Some parts required more elaborative definitions and some required paraphrasing. Although there are many studies have been done on GRF genes of aquatic plants, a comprehensive characterization of the gene expression level between species under certain trigger is relatively new. I suggest the authors to take a major revision to improve the clarity of the content.

General Overview

When I read the abstract as someone who just comes by, reading the first sentence, I can get the good premise of this study. However, I became confused when I read "...floating plants, floating-leaved plants, emergent plants, submersed plants and amphibious plants." Could you please add the species in the parentheses (i.e., floating plants (Spirodela polyrhiza))? Many researchers scroll around the abstract to see if a paper match their interests, especially at the species names. Adding the species name after the plant selection types can attract the audience more. I think by adding these details, there are still plenty room for the abstract. 

Referring to Spirodela polyrhiza (the typical floating plant) [line 21; Abstract; line 61; Introduction]. Could you use the common name? "Typical floating plant" is not typical everywhere. In some other countries, a typical floating plant could be Potenderia crassipes or water hyacinth, some could have Pistia stratiotes or water lettuce. I believe you can refer S. polyrhiza as duckweed, duckmeat, or if you want to add a touch of "local wisdom" here, you can write it like this: duckweed (Spirodela polyrhiza) or also known locally at <your place> as <local name>.

Of all phytohormones, why do you specifically choose ABA? Why not auxin (either natural or synthetic - especially since the synthetic auxin could potentially harm the plant), cytokinin, GA, BR, JA, SA? I wish to see the strong reasoning for this in the introduction section, because I didn't see the weight of why the authors pointed to ABA.

Introduction

Line 30-33: Is someone has reported the type of NLS belongs to QLQ and WRC? Are the NLS monopartite or bipartite? If not, perhaps you can add it to your experiment here. All you have to do is add the protein sequence to a NLS detecting web and they will immediately show you which sequence is the NLS and the type. Perhaps you want to see NLS Mapper (https://nls-mapper.iab.keio.ac.jp/cgi-bin/NLS_Mapper_form.cgi). In term of bioinformatics, I think more data to describe, will be better.

Line 47-49: Be careful, not all of them. Many aquatic oxygen suppliers are submerged plants, I believe. As following the photosynthesis, their oxygen will be released to the water. The floating plants, on the other hand, their stomata are often located on the leaf, which stomata are located on the adaxial surface. Hence, their oxygen are released to the atmosphere, not to the water. On unrelated note, don't you think aquatic plants are good metal accumulator that could help bioremediation?

Line 63-64: You finally referred "duckweed" here, but you didn't elaborate what "duckweed" is before. That's why previously I strongly suggest to describe the common name of S. polyrhiza first. 

Results

Line 133-134: Referring to "N. nucifera (Nelumbo), N. colorata (Nymphaea), C. chinensis (Cladopus) and A. thaliana (Brassicaceae)", what are the implication of the clade name inside the parentheses? Almost all are genus, but for A. thaliana refers to the family name. Also, perhaps you can use different genus abbreviation between Nelumbo and Nymphaea to avoid confusion, for instance: Ne. nuciferaNy. colorata. Not only here, but throughout the manuscript.

Figure 2 and 3. Could you elaborate the meaning of colors in the figure belong to which species in figure descriptions? Specifically for Figure 3, the figure legends are too small.

Materials and Methods

Line 245-261: Please correct the scientific binomial writing (Italic, capitalization)

I want a clarification here: So you obtained the genome database, is it as a whole genome database or whole protein database? If genome, how do you perform BLASTp? BLASTp is meant to search protein on protein database. Don't we commonly use tBLASTn (to search protein on translated nucleotide database)?

Line 255: If we see into Pfam website (http://pfam.xfam.org/) the website is about to be powered down in October. If you don't mind for reproducibility reason, also do on either or both HMMER (https://www.ebi.ac.uk/Tools/hmmer/search/phmmer) and InterPro (https://www.ebi.ac.uk/interpro/search/sequence/). As I mentioned before, perhaps you want to identify the NLS in the NLS Mapper (https://nls-mapper.iab.keio.ac.jp/cgi-bin/NLS_Mapper_form.cgi) as well. From these, you can have more comprehensive data to explain in the results.

Reviewer 2 Report

Dear Authors,

I had agreat oportunity to review manuscript entidled: ” Genome wide analysis of the growth regulating factor (GRF) family in aquatic plants and their roles in the ABA-induced turion formation of Spirodela polyrhiza” which is considered for publication in IJMS journal. Article presents generally interesting new insights in genome wide analysis of Spirodela sp. But in some points need a improvements which I present in a form of points below:

Minor problems:

1.       Introduction section

According IJMS publication rules this part of manuscript should eneded with aim of the study and/or hypothesis. Currently this section did not have any precisely presented aim of the study.

2.       Results

Figure 2 is too small and low quality to be read and understand properly

Figure 3 is extremely low quality and to complex I recommend to split this figure into two Figures in HD quality

Figure 4 as sam as Figure 3

Figure 5 A and B parts are too low quality I suggest to split this figure to two separate figures one with parts A and B and Second with part C

Figure 6 all elements of Spirodela must be marked on photo the phenotype must be also named exacly in text of results or in Figure decription. Morover statistic analys on part B of this figure is extremely strange because almost all values are not statistically significant moreover the values for SpGRF5 are marked by authors as significance different but truly the charts (also marked confidence intervals). I strongly recommend recalculation based of ANOVA and mark by use letters all significance or lack of significance on expression

Part A of Figure 7 is too small and too low quality

The supplementary materials as tables should be rather in pdf format not exel because TAbleS1 in Excel is marked as Sheet 1 not truly table.

3.       Materials and methods

5.2. Physiochemical Properties and Subcellular Localization this is error I suggest to change this as predicted putative localization. In the results section I suggest also to use term predicted putative localization because it is only

Sincerely,

Round 2

Reviewer 1 Report

I would like to thank the authors for making such extensive revisions for the manuscript. The level of comprehensiveness of the manuscript has been greatly improved. Only few requests from me remain.

You have done well on performing the NLS prediction for the GRFs, as I clearly noticed the edits in the results. Curious, are there references indicating which type of NLS are present in the other GRFs (does not have to refer on the species that you study here, but in other reference species)? Are the GRFs can have either monopartite or bipartite NLS and not very specific to only one type? If so, put them to the discussion section. If none, you can highlight your discovery of these NLS details in the discussion section, too. You can discuss the functionality or even from the evolutionary perspective (phylogenetic, etc.). Does not have to be a paragraph-long discussion, but you can make it briefly (so it won't be off focus).

For method section (Line 309-312), please crosscheck the link addresses, and did you mean InterProScan instead of InterPro? Please correct me if I'm wrong. For HMMER, you can just refer to the general/main web: http://hmmer.org. Unless you are referring that you used "PHMMER feature of HMMER (https://www.ebi.ac.uk/Tools/hmmer/search/phmmer)" where you can use it to input your protein database to get it compared to the protein sequence database. If so, please specifically mention which database that you use (e.g., Reference Proteomes, UniProtKB, etc.).

Reviewer 2 Report

Dear Authors,

All improvements is added. I recomend publication

Sincerely,

Author Response

Dear reviewer, thank you very much for your help.